# "Mistakes Help Us Grow": Facilitating and Evaluating Growth Mindset Supportive Language in Classrooms

**Kunal Handa**[1]    **Margaret Clapper**[2]    **Jessica Boyle**[3]    **Rose E Wang**[4]

**Diyi Yang**[4]    **David S Yeager**[2]    **Dorottya Demszky**[4]

[1]Brown University    [2]UT Austin    [3]Vanderbilt University    [4]Stanford University

kunal_handa@alumni.brown.edu
ddemszky@stanford.edu

## Abstract

Teachers' growth mindset supportive language (GMSL)—rhetoric emphasizing that one's skills can be improved over time—has been shown to significantly reduce disparities in academic achievement and enhance students' learning outcomes. Although teachers espouse growth mindset principles, most find it difficult to adopt GMSL in their practice due the lack of effective coaching in this area. We explore whether large language models (LLMs) can provide automated, personalized coaching to support teachers' use of GMSL. We establish an effective coaching tool to reframe unsupportive utterances to GMSL by developing (i) a parallel dataset containing GMSL-trained teacher reframings of unsupportive statements with an accompanying annotation guide, (ii) a GMSL prompt framework to revise teachers' unsupportive language, and (iii) an evaluation framework grounded in psychological theory for evaluating GMSL with the help of students and teachers.[1] We conduct a large-scale evaluation involving 174 teachers and 1,006 students, finding that both teachers and students perceive GMSL-trained teacher and model reframings as more effective in fostering a growth mindset and promoting challenge-seeking behavior, among other benefits. We also find that model-generated reframings outperform those from the GMSL-trained teachers. These results show promise for harnessing LLMs to provide automated GMSL feedback for teachers and, more broadly, LLMs' potentiality for supporting students' learning in the classroom. Our findings also demonstrate the benefit of large-scale human evaluations when applying LLMs in educational domains.

## 1 Introduction

Growth mindset supportive language (GMSL) refers to rhetoric highlighting the potential for individuals to develop their skills and abilities over time. Research on GMSL has shown positive effects on crucial aspects of student learning, such as resilience, motivation, and performance (Dweck, 2006). This language also helps reduce educational gaps linked to race, ethnicity, and social class (Hecht et al., 2023; Canning et al., 2019).

Despite many teachers embracing the importance of GMSL, most find it difficult to implement (Dweck and Yeager, 2019) due to misconceptions as to what constitutes a growth mindset and the lack of effective coaching in this domain. Prior work has shown that many well-intentioned teachers often believe they are using GSML (for example, through simply adding more positive expressions—"I believe in you"), without comprehensively understanding the tenets of GMSL needed to apply it in practice (Dweck and Yeager, 2019; Dweck, 2016). Helping teachers adopt growth mindset principles requires coaching and individualized feedback that is currently not available at scale (Hecht et al., 2023).

Recent advancements in Large Language Models (LLMs) have led to unparalleled abilities for models to follow instructions, which suggests potential for prompting them with growth mindset principles to generate suggestions for teachers. Although unexplored with GMSL, LLMs have shown success as automated educational tools in other settings. For example, KhanMigo (Khan Academy, 2023), an LLM-assisted tutoring bot was successfully prompt-engineered to follow principles of Socratic questioning and not provide the answer to students directly. At the same time, recent work has also shown that ChatGPT — a model from the same family as KhanMigo — was not successful at providing teachers with high-level, insightful pedagogical feedback (Wang and Demszky, 2023). Thus, the question whether LLMs are capable of generating GMSL for teachers remains an open question.

In this paper, we seek to assess the ability of

---

[1]https://github.com/kunhanda/growth_mindset

GPT-4, a state-of-the-art LLM, to generate GMSL. In doing so, we make several contributions. First, working with teachers trained in growth mindset principles (henceforth referred to as "expert teachers"), we develop an **annotation framework** to guide teachers' understanding of GMSL. Leveraging a dataset of elementary math classroom transcripts (Demszky and Hill, 2023), we further collaborate with these teachers to create an open-source **parallel dataset** that pairs 100 unsupportive utterances from the transcripts with their expert-revised GMSL reframings.

Having outlined the tenets of GMSL, we **prompt engineer GPT-4** (OpenAI, 2023) to generate GMSL reframings of the unsupportive utterances. We utilize recent advancements in prompting and self-critique (Huang et al., 2022) techniques to design a detailed, GMSL-specific prompt. Via our prompt design, we ensure that the reframings are personalized for each specific classroom instance and adhere to our GMSL tenets.

Critically, we also propose a novel **framework for evaluating GMSL**, assessing teacher and model responses by recruiting 174 teachers and 1,006 students (13-15 years old) to evaluate the perceived impact that these responses would have on students. Our participants are given excerpts from a conversation between a teacher and student and asked to rate the teachers on their (1) perceived growth mindset, (2) promotion of challenge-taking behavior, (3) alleviation of students' feelings of shame, and (4) increase of respect felt by students. Some excerpts contain the original teacher responses whereas others contain reframings (either by the expert teachers or model). We adapt these questions from instruments developed and tested by growth mindset experts (Hecht et al., 2022).

In our evaluation, we find that both expert teachers' and model reframings better support growth mindset, outperforming the original utterances across all metrics. We discover that model reframings slightly surpass even the expert teachers' reframings and that the model with self-critique performs best. We also conduct a lexical analysis, identifying that all reframings include more explicit growth mindset supportive language than the original (e.g. empathetic validation, positioning oneself as a collaborative resource, autonomy supportive language), with models including even more specific diction compared to teacher utterances.

Our findings not only highlight the need for encouraging GMSL in classrooms but also offer valuable insight into the role LLMs can provide in this endeavor. Furthermore, our results highlight the importance of extensive student-centered evaluations when examining the usefulness of LLMs in educational contexts, offering a framework for future studies in this area. We demonstrate that through closely working with students and teachers when developing and evaluating LLM-based technologies, we can help ensure their positive influence within the classroom.

## 2 Related Works

### 2.1 Growth Mindset Supportive Language

Growth mindset refers to the belief that intelligence and abilities are not fixed and can be developed over time (Dweck, 2006; Yeager et al., 2016). This perspective emphasizes the potential for individuals to improve their skills and intelligence through effort, learning strategies, and overcoming challenges.

Previous studies have revealed that incorporating GMSL positively and lastingly influences students' academic performance; for example, Yeager et al. (2019) found that exposing students to a growth mindset in a short one-hour conversation improved the grades among low-achieving students and increased enrollment to advanced mathematics courses (see also Schmidt et al. (2016) and Muenks et al. (2020) for similar findings). Results were consistent across achievement levels and countries (OECD, 2019). Additional research has also shown that a growth mindset can positively impact students' engagement, persistence, and resilience in the face of academic challenges (Good et al., 2003; Yeager et al., 2014; Dweck and Yeager, 2019). In contrast, when an educator does not foster and support a student's mindset with their own growth mindset language and behaviors, students are less likely to act on their growth mindset beliefs (Yeager et al., 2021). Furthermore, recent work has also validated the use of survey-based evaluation setups when judging the long-term impact of GMSL intervention; Hecht et al. (2022) found that students' hypothetical judgments of teachers' mindset supportive language track with teachers' actual behavior and students' learning outcomes.

These prior works highlight the importance of using GMSL in classrooms. Yet, providing feedback that encourages educators to incorporate this growth-mindset-oriented feedback can be incredi-

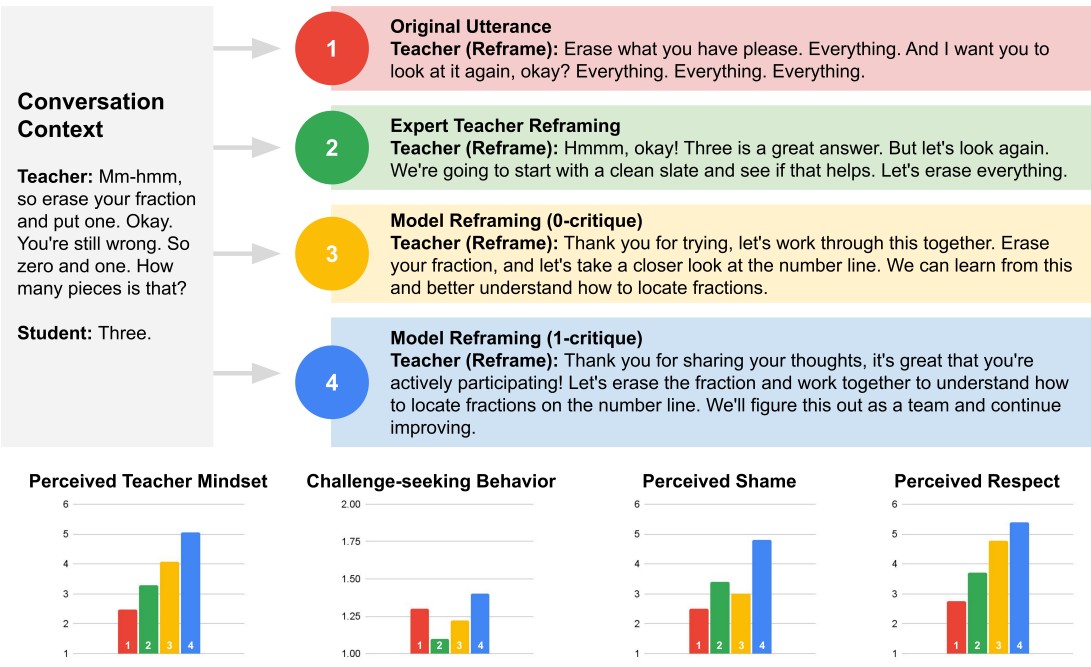

Figure 1: Examples from our dataset illustrating different responses to a student's mistake by teachers (Original and Expert) and GPT-4 (with and without self-critique). The figure also shows students' ratings of these responses along four dimensions from our evaluation framework: Perceived Teacher Mindset, Challenge-Seeking Behavior (choice of easy vs hard assignments), Perceived Shame, and Perceived Respect.

bly difficult to do at scale. Hunkins et al. (2022) focus on detecting GMSL but do not explore reframing unsupportive language at scale. Recent work like Ziems et al. (2022) explores the use of positive reframing to generate growth-mindset-oriented language. However, their study relies on social media data, which requires a different approach to reframing compared to our educational domain.

Our research builds upon the demonstrated beneficial impacts of GMSL on students' long-term learning outcomes. We aim to explore the utility of LLMs as an effective and individualized method for coaching teachers to adopt growth mindset principles through a large-scale survey-based evaluation framework.

## 2.2 Automated Feedback to Educators

Prior works on automated feedback tools provide analytics on student engagement and progress (Su et al., 2014; Schwarz et al., 2018; Aslan et al., 2019; Bonneton-Botté et al., 2020; Alrajhi et al., 2021, among others). These tools enable teachers to monitor student learning and intervene as needed. Recent NLP advances are able to provide teachers feedback on their classroom discourse, promoting self-reflection and instructional development (Samei et al., 2014; Donnelly et al., 2017;

Kelly et al., 2018; Jensen et al., 2020; Wang and Demszky, 2023). For example, Suresh et al. (2021) provides feedback to teachers on their teaching moves, such as how frequently the teacher revoices a student's idea or how frequently the teacher asks students to reason aloud. Jacobs et al. (2022) provides evidence that K-12 math teachers receive this kind of feedback positively. A similar tool, M-Powering Teachers, provides feedback to teachers on their uptake of student ideas and demonstrates effectiveness in the 1-on-1 learning setting (Demszky and Liu, 2023) and online group instruction (Demszky et al., 2023). Altogether, these findings show a positive impact of cost-effective automated tools in educational contexts.

## 3 Modeling GMSL

We utilize a dataset of elementary math classroom transcripts (Demszky and Hill, 2023) and GPT-4 prompting techniques to create four different types of responses:

1. ORIGINAL: Teacher utterance from the classroom transcript labeled as unsupportive by teachers trained in GMSL.

2. EXPERT: Reframings of unsupportive examples to GMSL by teachers trained in GMSL.

**System Text**
You are a helpful assistant who specializes in correcting teachers' language to be more supportive of the students. Your task is to directly revise the teacher's utterance to use mindset-supportive language. Below are some tenets of mindset-supportive language…
1. Empathic validation…
2. Reappraisal of affect…
3. Seek to Understand...
4. Position yourself as a collaborative resource…
5. Provide hope for change in the future…
6. Autonomy-supportive language throughout…

**Prompt Text**
This is an elementary grade math class working on division. They use manipulatives to build arrays and separate the manipulative into equal groups.
The following example of an in-class conversation between a teacher and student contains unsupportive language. Directly revise the teacher's last utterance, marked with "(Reframe)", to be more mindset-supportive:

**Teacher:** Mm-hmm, so erase your fraction and put one. Okay. You're still wrong. So zero and one. How many pieces is that?

**Student:** Three.

**Teacher (Reframe):** Erase what you have please. Everything. And I want you to look at it again, okay? Everything. Everything. Everything.

**Model Reframing (0-critique)**
**Teacher (Reframe):** Thank you for trying that solution. Let's erase your current answer, and together we can take another look at the problem and consider other ways to approach it. I believe we can figure this out as a team.

**Prompt:**
Make the previous response more mindset supportive.

**Model Reframing (1-critique)**
**Teacher (Reframe):** I appreciate your effort in attempting that solution. Let's gently erase what you've written and revisit the problem together. We can explore different approaches to better understand it. Remember, we're in this together and can learn from each attempt.

Figure 2: Our prompt design employed to elicit GMSL reframings using GPT-4 as described in Section 3.2.

3. MODEL (0-CRITIQUES): Reframings of unsupportive examples to GMSL by GPT-4.

4. MODEL (1-CRITIQUE): Reframings of unsupportive examples to GMSL by GPT-4 prompted to self-improve upon responses.

The subsequent sections detail the development of these four types of responses.

## 3.1 Data

**Source.** We source our data from the National Center for Teacher Effectiveness (NCTE) 4-5th grade math classroom transcripts (Demszky and Hill, 2023), representing 317 teachers across four school districts in New England, predominately serving low-income students of color. Transcripts are anonymized, where student and teacher names are replaced with terms like "Student A" and "Teacher B".

**Expert teachers.** We recruit two teachers who are alumni of the Hollyhock Fellowship Program[2], a competitive two-year teacher professional learning program that works with highly motivated teachers serving in Title I schools. Both teachers

have 8+ years of experience and have experience teaching elementary grade levels. Both teachers are women; one of them identifies as Hispanic and the other as white. In addition to the training these teachers had already received in growth mindset, our team provided the teachers additional guidance for GMSL principles. We work closely with the teachers and compensate them at $50/hr.

**Identifying unsupportive utterances.** To compile a dataset of unsupportive teacher utterances (ORIGINAL), we ask expert teachers to annotate all teacher utterances in a random sample of transcripts for whether they (I) respond to a student mistake, introduce a task or debrief a task and (II) if so, whether they contain unsupportive language as defined by the annotation guide described below. We follow this process until we reach 100 examples of unsupportive teacher utterances. These examples are derived from eight transcripts and 100 total teacher utterances meeting criterion (I), indicating a high rate (22.8%) of unsupportive language in the transcripts.

**Reframing.** We worked closely with the expert teachers to create a GMSL Guide that documents their process for reframing teacher utterances to

[2]https://cset.stanford.edu/pd/hollyhock

be mindset supportive (EXPERT). The Guide incorporates suggestions for teachers on fostering a growth mindset in classrooms. These suggestions include providing empathetic validation by expressing gratitude to students for asking questions and demonstrating effort, positioning oneself as a collaborative resource by utilizing inclusive terms like "we" and "us" when engaging in problem-solving activities with students, and seeking to understand students' thinking by asking thoughtful questions. After multiple rounds of pilot annotations and feedback sessions, the two expert math teachers utilized the Guide to independently identify examples of unsupportive language and subsequently reframe them to foster a more supportive mindset while preserving the task-related meaning of the message (a more detailed description of the process can be found in the Appendix, Section A).

## 3.2 Designing a Prompt to Elicit GMSL

We design a GMSL-oriented prompt to generate reframings of teachers' unsupportive utterances. We first engineer GMSL-specific information given to the model via its "system text"—high-level instructions that can contain extra content to guide the model's responses (Microsoft, 2023).

Here, the model receives both instructions specifying the model's role in correcting teachers' language to be more supportive of students as well as the GMSL Guide, outlining the tenets of GMSL. When prompting the model, we provide the model with (1) the classroom context in which the conversation to be reframed took place (e.g. *This is an elementary grade math class working on geometry. They start the lesson with reviewing multiplication facts. They then begin to learn about classifying triangles focusing on types of angles.*), (2) explicit instructions of the task: *The following example of an in-class conversation between a teacher and student contains unsupportive language. Directly revise the teacher's last utterance, marked with "(Reframe)", to be more mindset supportive*, and (3) the utterance requiring reframing, demarcated by "*(Reframe)*", preceded by the two utterances directly prior. We use the model's generated response in our evaluation (MODEL (0-CRITIQUES)). This setup closely follows that of the expert teacher annotators; but, due to context-window limits, we were unable to provide the entire conversation leading up to the utterance, as was seen by the expert teachers.

Recent advances in prompting techniques have demonstrated the value of self-critique in LLMs—prompting the model to improve upon its own response—enabling better reasoning about tasks (Huang et al., 2022). We utilize this process of self-critique when reframing, evaluating the model generations with one self-critique and zero self-critiques. For generations with one self-critique, we prompt the model to improve upon its initial response (*Make the previous response more mindset supportive*), evaluating only the final generation (MODEL (1-CRITIQUE)).

## 4 Evaluation

We develop a framework, grounded in psychological theory, for evaluating GMSL in generated responses with the help of teachers and students (Section 4.1). In addition, we also conduct a lexical analysis to understand linguistic differences among the different types of responses (Section 4.2).

### 4.1 Teacher and Student Surveys

In NLP, human evaluations of text generation tend to focus on rating the *quality* of generated text rather than the *impact* the text may have on the target recipient. In social psychology, impact evaluations are paramount, as researchers seek to measure whether their intervention had the desired effect on their participants. Furthermore, many psychological constructs such as growth mindset are complex and subjective and hence they cannot be evaluated directly — e.g. asking raters "How much growth mindset does this text convey?" is likely to led to confusion and unreliable annotations.

**Measures of GMSL.** To obtain measurements of GMSL, we adapt survey items validated as part of large-scale growth mindset interventions (Hecht et al., 2022). We administer surveys to both teachers' and students' to probe their perceptions of the classroom and the teacher. The surveys include items estimating four constructs given an excerpt of a student-teacher interaction: (1) teacher's **perceived growth mindset**, (2) teacher's promotion of **challenge-taking behavior**, (3) **shame** felt by students in the teacher's class, and (4) **respect** felt by students in the teacher's class. These constructs operationalize growth mindset culture and were designed and tested by experts (Hecht et al., 2022).

Figure 3 shows the key items on the student survey corresponding to the four aforementioned constructs. The teacher survey included largely

```
BASELINE MOTIVATIONAL FRAMEWORK (control variable)
b1 - To be honest, you can't really change how intelligent you are.
b2 - You can always substantially change how intelligent you are.
b3 - When you try really hard on a subject in school, it means you can't really be good at that
subject.
PERCEIVED TEACHER MINDSET (outcome variable)
ptm1 (r) - This math teacher seems to believe that only some students will understand the hardest
problems.
ptm2 (r) - This math teacher seems to believe that student's can't really change how good they
are at math.
SHAME (outcome variable)
shame (r) - If I were in this teacher's class, I would feel embarrassed if I got a problem wrong
on the board in front of my peers.
RESPECT (outcome variable)
respect - In this class, the teacher would treat me with respect.
CHALLENGE-SEEKING BEHAVIOR (outcome variable)
csb - Imagine that, later today or tomorrow, this math teacher handed out two extra credit
assignments. You got to choose which one to do. You get the same number of points for trying
either one. One choice is an easy review—it has math problems you already know how to solve,
and you will probably get most of the answers right without having to think very much. It takes
30 minutes. The other choice is a hard challenge—it has math problems you don't know how to
solve, and you will probably get most of the problems wrong, but you might learn something new.
It also takes 30 minutes. If you had to pick right now, which would you pick?
```

Figure 3: Items used in the survey evaluations for adolescents. (r) indicates a reverse coded item, where higher scores are "worse".

the same items as the student survey (except csb), with minor wording adaptations. The shame and respect items were adapted so that there were not from the student's point of view. Instead, teachers answered the following items: *"Students in this teacher's class would probably feel embarrassed if they got a problem wrong on the board in front of their peers."* and *"In this class, the teacher would treat students with respect."* Only students were asked to respond to a hypothetical challenge-taking item (csb) in which they imagine an opportunity to receive extra-credit in the teacher's class. When students perceive the fictional teacher as having more of a fixed mindset, they are more likely to accept the easy extra-credit assignment, regardless of the student's own baseline growth mindset beliefs.

All items in Figure 3 except csb were rated on a 1-6 Likert scale with the choices being *Strongly Disagree, Disagree, Mostly Disagree, Mostly Agree, Agree, and Strongly Agree*. The csb item was rated on a binary scale, indicating if a student would choose easy vs hard assignments.

**Survey design & flow.** The full teacher and student surveys are included among the Supplementary Materials. After the consent form, the survey first asked participants about their baseline growth mindset beliefs. These items are generally referred to as participants' **baseline motivational frame-**

**work** and consist of items b1, b2, and b3 in Figure 3. We use responses to these items as control variables in our analyses.

Then, they were shown four transcript segments one by one. Each segment included a different response type (ORIGINAL, EXPERT, MODEL (0-CRITIQUES), MODEL (1-CRITIQUE)). The segments the participants saw and the order they viewed them in were randomized. Participants were not told to directly compare the segments and generally did not see reframed utterances from the same segment. Following each segment, participants were asked to respond to the GMSL items described above (ptm1, ptm2, shame, respect and for students, csb). These measures are used in later analyses as outcome variables. Finally, consistent with standard psychology practices, we ask all participants about their demographics. Participants responded to items about their self-identified gender, race and ethnicity, and mother's highest level of education (a socio-economic measure). At the end of the survey, after responding to all measures, participants were told that some of the transcripts they viewed were edited by artificial intelligence.

**Participants.** We recruit 174 teachers and 1,006 students as evaluators. As for teachers, we recruit them from two pools: 16 teachers are alumni of the Hollyhock Fellowship Program, the same pro-

fessional learning program that we recruited our two expert teachers from (the two experts are not included in this pool). The other 158 teachers are recruited via Prolific (Palan and Schitter, 2017). We recruit students through Dynata[3], a service that helps recruit participants for surveys meeting specified demographic criteria. The youngest population they are able to recruit is 13-15, and we work with this group. Our study is approved under IRB approval number: HRP-UT902 STUDY00001380.

**Analysis of survey results.** We assess the extent to which the four types of model responses contain GMSL by analyzing teacher and student perceptions collected through the survey. Following standard procedures in psychology, we control for raters' demographic features and their baseline motivational framework when estimating average ratings for each model response. We use the following model to estimate these conditional means:

$$y = \beta_0 + D\beta_1 + B\beta_2 + \varepsilon \qquad (1)$$

where $y$ is a vector representing the outcome, $D$ is a matrix of rater demographic characteristics, $B$ is a matrix of responses to baseline motivational framework items, $\beta_0$, $\beta_1$ and $\beta_2$ are vectors of unknown parameters to be estimated and $\epsilon$ is a vector of residuals. We report the conditional mean computed when all variables $D$ and $B$ are at their mean.

The demographic features $D$ include binary indicators for the rater identifying as female, Asian, Black/African American, White/Caucasian, Hispanic/Latinx, Pacific Islander/Hawaiian, dummy coded variables for mother's education level and for teachers, whether they were part of the Prolific pool. For the outcomes $y$ and motivational framework $B$, we use the numerical Likert scale responses. To obtain a single outcome for perceived teacher mindset, we first reverse the scales for `ptm1` and `ptm2` (so that greater values reflect more growth mindset) and average ratings for the two items. We keep the other responses as is.

### 4.2 Lexical Analysis

In addition to surveys, we also conduct lexical analysis to understand the linguistic differences among different response types. These observations can contribute to a qualitative explanation of student and teacher evaluations regarding the language that nurtures a growth mindset classroom

[3]https://www.dynata.com/

environment. For example, GMSL keywords such as "challenging" and "together," or first-person pronouns like "we" and "us," might be more prevalent in responses perceived to be said by a teacher with a growth mindset.

We compute the log odds ratio, latent Dirichlet prior, measure defined in (Monroe et al., 2008) to estimate the distinctiveness of an n-gram appearing in a particular response type (ORIGINAL, EXPERT, MODEL (0-CRITIQUES), or MODEL (1-CRITIQUE)) as opposed to all other response types. The log odds estimate is normalized to standard deviation units (e.g. a score of 2 can be interpreted as the n-gram being 2 standard deviations "more associated" with a particular response type than others). We preprocess the data using Python's NLTK package (Bird et al., 2009) for tokenization, lowercasing, and lemmatization, while also discarding stop words and non-alphanumeric tokens. We use the Gensim Phrases Python package (Rehurek and Sojka, 2011) to retrieve unigrams and frequent bigrams in the dataset.

## 5 Results

### 5.1 Teacher and Student Surveys

Table 1 shows the results of the student and teacher evaluations. One high-level take-away is that ratings for the different items corroborate one another, surfacing a systematic ranking among the different response types in terms of GMSL: ORIGINAL < EXPERT < MODEL (0-CRITIQUES) < MODEL (1-CRITIQUE). We find the model responses outperform the ORIGINAL teacher response by a large margin along all dimensions, with MODEL (1-CRITIQUE) doing 24-85% better and MODEL (0-CRITIQUES) doing 23-75% better. Notably, MODEL (1-CRITIQUE) almost doubles students' likelihood of choosing hard assignments (hypothetical challenge-seeking behavior), which is the most important GMSL student outcome, in contrast to ORIGINAL. As expected, EXPERT also scores better (by 17-50%) in terms of GMSL than the ORIGINAL. However, unexpectedly, the models also consistently receive better ratings than the EXPERT teacher response, showing 8-10% increased perceived growth mindset, 9-15% decreased shame, 9-10% increased respect, and importantly, 23% increased challenge-seeking behavior. We also find that self-critique facilitates GMSL, increasing ratings for MODEL (1-CRITIQUE) compared to MODEL (0-CRITIQUES) by 1-6%. These re-

| | Teacher Evaluation | | | Student Evaluation | | | |
|---|---|---|---|---|---|---|---|
| Response Type | Perceived Teacher Mindset | Shame | Respect | Perceived Teacher Mindset | Shame | Respect | Challenge-Seeking |
| ORIGINAL | 3.42 | 4.15 | 3.77 | 3.3 | 4.4 | 3.6 | 0.2 |
| EXPERT | 4.38 | 3.11 | 4.73 | 3.94 | 3.67 | 4.47 | 0.3 |
| MODEL (0-CRITIQUES) | 4.69 | 2.7 | 5.03 | 4.14 | 3.39 | 4.78 | 0.35 |
| MODEL (1-CRITIQUE) | **4.8** | **2.63** | **5.19** | **4.25** | **3.34** | **4.89** | **0.37** |

Table 1: Conditional means computed based on teacher (n=174) and student (n=1,006) survey responses. The values show the average ratings of the four response types along our key outcome dimensions, while controlling for participant's demographics and baseline motivational framework. Results indicate that the model responses are consistently rated as more growth mindset supportive compared to the original and even the expert teacher responses, with MODEL (1-CRITIQUE) performing slightly better than MODEL (0-CRITIQUES).

| ORIGINAL | | EXPERT | | MODEL (0-CRITIQUES) | | MODEL (1-CRITIQUE) | |
|---|---|---|---|---|---|---|---|
| N-grams | Log Odds | N-grams | Log Odds | N-grams | Log Odds | N-grams | Log Odds |
| guy | 6.40 | tool | 6.23 | let_work | 7.08 | teacher_appreciate | 6.72 |
| let_see | 5.95 | let_look | 6.23 | keep_mind | 5.06 | let_collaborate | 6.38 |
| sit | 5.80 | teacher_oh | 6.05 | giving_try | 5.06 | let_consider | 5.62 |
| everybody | 5.80 | let_go | 6.05 | appreciate_input | 5.06 | great_see | 5.33 |
| head | 5.80 | tell_u | 5.82 | find_right | 4.77 | explore_different | 5.33 |
| gon_na | 5.80 | complete | 5.82 | think_another | 4.77 | engaging_problem | 5.15 |
| teacher_erase | 5.61 | let_talk | 5.54 | teacher_alright | 4.77 | let_focus | 5.15 |
| supposed | 5.39 | hmmm | 5.54 | understand_better | 4.77 | learn_grow | 5.15 |
| right_think | 5.39 | great_answer | 5.54 | working_together | 4.77 | continue_exploring | 4.93 |
| go_back | 5.39 | toolbox | 5.54 | way_approach | 4.77 | great_effort | 4.93 |

Table 2: The 10 n-grams with the highest log odds for each type of response. EXPERT utilizes more GMSL (as defined in the GMSL Guide) compared to ORIGINAL. MODEL (0-CRITIQUES) and MODEL (1-CRITIQUE) include more GMSL than the EXPERT.

sults suggest that GPT-4 is able to successfully incorporate and self-improve along growth mindset principles given effective prompts.

## 5.2 Lexical Variation Among Responses

Our linguistic analysis highlights differences in word usage across the four response types. Table 2 presents the top 10 n-grams ordered by their log odds ratio. In comparison to ORIGINAL, EXPERT responses contain more pronounced examples of GMSL, such as empathic validation with phrases like *great answer* and inclusive expressions like *let's look* and *let's talk*. Both types of model reframings exhibit even more explicit GMSL, as evidenced by autonomy-supportive language (e.g., *let's consider*), positioning as a collaborative resource (e.g., *working together*, *let's collaborate*), hope for change (e.g., *continue exploring*), and empathic validation (e.g., *great effort* and *appreciate input*). These findings suggest that all reframings are more effective in using GMSL compared to original utterances, while model reframings are

especially adept at utilizing GMSL elements.

One noticeable pattern is that the models, despite having been provided with context for grade level, use more formal language (e.g. *appreciate, engage, collaborate*) than the teacher responses, which may be harder for younger students to understand. Thus, this lexical analysis highlights the need for additionally evaluating the grade-appropriateness of teacher language beyond GMSL and importance to not use LLMs directly with students without a teacher in the loop who can adapt language to the learners' contexts, needs, and background.

## 6 Discussion & Conclusion

In this study, we take steps towards helping teachers adopt growth mindset supportive language (GMSL) in their classroom. We establish an annotation guide, create a parallel dataset, develop an automated coaching tool utilizing GPT-4, and propose a framework grounded in psychological theory for evaluating GMSL's effectiveness for both teachers and students. Our findings demonstrate that

expert-teacher and model-generated reframings significantly improved upon the original unsupportive utterances in promoting a growth mindset, encouraging challenge-seeking behavior, alleviating feelings of shame, and enhancing the perceived respect by students. Interestingly, model reframings surpassed expert-teacher reframings, highlighting the potential of LLMs in fostering GMSL adoption within educational contexts.

Our work serves as a foundation for eliciting and evaluating the use of GMSL by teachers and LLMs through collaboration with students and teachers. The insights derived from our analysis open avenues for further inquiry into the role LLMs can play in the future to support teachers' communication strategies and facilitate better student-teacher interactions in the classroom. Future research could build upon this foundation to explore new dimensions of GMSL, examine the long-term impact of GMSL coaching, and extend the application of LLMs alongside our psychologically-grounded evaluation framework to address other outstanding challenges across various educational domains.

## Limitations

While this study offers valuable insights into the adoption of GMSL by teachers and the potential of LLMs in supporting educational communication, it is essential to acknowledge certain limitations. First, our analysis focuses on elementary math classroom transcripts, which may limit the generalizability of our findings for GMSL adoption across different subjects, age groups, or educational levels. Also, our study does not extensively address the potential cultural, linguistic, or regional differences in the understanding and interpretation of GMSL; we do not explicitly investigate the applicability of LLMs in this domain across all subject matters or classroom environments. Doing so remains necessary to build a broadly applicable framework for GMSL across diverse cultural and linguistic backgrounds. Additionally, our evaluations may not capture the complete breadth of students' characteristics, interactions, or the diverse spectrum of classroom situations needing different GMSL strategies. We also relied on teachers and students to evaluate the reframings based on hypothetical scenarios. This approach may not fully reflect the long-term impact of GMSL on student outcomes in real-life settings.

Future research could address these limitations by expanding the data sources to include varied subjects, age groups, cultural backgrounds, and classroom situations, as well as by studying the effects of GMSL across a broader set of contexts. Further research on LLMs' ability to produce GMSL across different teacher instruction styles, vernaculars, and personalities also remains necessary. Moreover, further exploration of the cultural and linguistic variability in GMSL implementation, alongside more extensive empirical studies on the long-term impact of GPT-4-based GMSL coaching, could contribute to a more comprehensive understanding of LLMs' utility in education settings.

## Ethics Statement

**Annotation.** The two teachers with whom we collaborated to develop the annotation guide and expert reframings were compensated at a rate of $50/hr. We met virtually with these teachers on a weekly basis and maintained available communication channels via email throughout the annotation process.

**Evaluation.** All participants were compensated above the federal minimum wage. Teachers recruited via Prolific were paid $13-20/hr, in line with the platform's guidelines. Teachers recruited for evaluation via the Hollyhock Fellowship Program were paid with $15 Amazon E-gift Cards (rate of $60/hr). During the evaluation process, we provided prompt responses to workers' inquiries (within 24 hours). Within the survey, we also used ethics practices standard for human-subjects research. Researchers who were able to access the data and recruit participants were trained on human-subjects research and certified via their institution. This project was approved by an institutional review board. Adults were always given informed consent (an example of the consent can be found in the Appendix, Section E) and were given the option to not consent to participate. For adolescents, parents were given an informed consent document and adolescents were also briefed on the potential harms (here, we estimated no harms greater than everyday life and discomfort due to boredom or fatigue) before continuing in the survey. Both parents and adolescents were given the option to not participate based on the consent form. Participants were not required to complete any of the questions to complete the survey, and were given the option to skip questions or to leave the survey entirely. For adolescents we collected aggregate

semi-identifiable data in the form of demographic data (such as gender, race/ethnicity, and mother's education level). Demographic data is important for evaluation and further analyses on this dataset, and cannot directly identify a participant. For adult participants recruited via the Hollyhock Fellowship Program, we collected demographic information and emails to process payment. Emails were removed from the dataset as soon as payments were completed. For the data collected via Prolific, we only collected demographic data.

**Deployment.** Our data, prompt design, experiments, and evaluations are intended solely for bettering student-teacher interactions and positively influencing students' learning outcomes. Although we took great care in designing our project with each of these aspects with this in mind, we recognize that there may be unintended perverse use cases of our research, such as altering prompts that oppose our outlined goals. We urge parties intending on using any part of our work to align their intentions with those of this research: supporting educational experiences for students. Any use cases that attempt to deploy this research for commercial gain are unacceptable. We implore individuals who use our data, prompts, experiments, or evaluations to consider and mitigate other societal, ethical, or otherwise deleterious ramifications that may arise.

## Acknowledgements

We would like to thank Alex Tamkin, Cameron Hecht, and Carol Dweck for useful conversations and feedback, as well as all the participants who took our study. We would also like to thank the Hollyhock Fellowship Program alumni with whom we worked to create the GMSL Guide. R.E.W is supported by the National Science Foundation Graduate Research Fellowship. This work was funded partially from an NSF IIS-2247357 to D.Y. Additionally, this work was supported by research grants to the Texas Behavioral Science and Policy Institute, including the National Science Foundation under award numbers 1761179 and 2201928 (PI: D.S.Y.), by the National Institutes of Health under award numbers R01HD084772 (PI: D.S.Y.) and P2CHD042849 (Population Research Center), and by the William and Melinda Gates Foundation under awards INV-047751 and INV-004519 (PI: D.S.Y.). This work was also supported by an Advanced Research Fellowship from the Jacobs Foundation to D.S.Y.

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

## Overview of Appendix

We provide additional information on the development of the GMSL Guide, prompt designing, prompt text, and survey creation. We also provide a diverse set of examples of model and teacher reframings in five different conversations as well as evaluators' ratings of the original utterances and reframings for each conversation.

Our code is available at https://github.com/kunhanda/growth_mindset. Due to our IRB agreement, we are not permitted to release the survey data without having individuals sign a data-sharing agreement. If you would like to access either the student or teacher survey data, please contact kunal_handa@alumni.brown.edu.

## A Developing the GSML Guide

The GMSL Guide described in Section 3.1 is developed in collaboration with growth mindset experts and elementary school teachers trained in GMSL. The process was as follows:

1. GMSL experts created an initial draft of a GMSL guide. This initial draft was informed by prior research highlighting the importance of educators' usage of GMSL (Yeager et al., 2021). The GMSL experts then met with the teachers to review the guide, review additional resources on general growth mindset information, and to discuss the internal thought process of identifying unsupportive language and how to reframe those instances into GMSL.

2. The teachers were then provided three practice NCTE transcripts and were asked to independently identify utterances needing reframing and to reframe those into GMSL. Anecdotally, many of the utterances selected for reframing were the same between teachers. These initial three transcripts were not used in our test set. During this phase, the teachers were also asked to document their annotation process.

3. Then, the GMSL experts met with the teachers to discuss their annotation process and GMSL reframing. In these meetings, we synthesized their experience of the annotation process to create an updated version of the GMSL Guide. During these conversations, we did not notice significant disagreement between the teachers.

4. After developing the more finalized version of the GMSL guide, the teachers then independently annotated the eight transcripts that were used in the test set. These final reframings were the ones used to prompt-tune and test GPT-4.

## B Iterating on Prompt Design

In designing our prompts to best elicit GMSL-specific language, we tested multiple different prompting strategies and the robustness of model responses.

### B.1 What other prompts did we test?

Our final prompts excelled in evoking GMSL from GPT-4 but earlier versions of the prompts did not perform as well. We tried a variety of different

prompts and prompting techniques before settling on the current prompt design described in the main text. We first attempted to categorize instances of unsupportive language in our dataset and provide the explicit categorization of the current example needing reframing when prompting the model. We found that this caused very predictable and robotic behavior from GPT-4. The model often outputted the same keywords used in the category (e.g. "Debriefing Task") in its generation.

We also tested prompting the model via in-context few-shot learning but found that the quality of generations significantly decreased. We hypothesize that this is due to the challenge of extracting the broad, salient aspects of GMSL in reframings. The model often perceived too context-specific aspects of the few-shot examples (e.g. generating a reframing mirroring that of one of its examples even if the conversational context was distinctly unrelated), ignoring the broad tenets of GMSL provided in the system text. In these cases, the model also often referenced the task teachers were discussing from the few-shot examples rather than applying the GMSL utilized in those examples to the task at hand.

Once we found that self-critique improved model performance in some cases, we attempted multiple different kinds of self-critique. We tried asking the model to improve upon its response to be better aligned with a specific GMSL tenet (e.g. *Improve upon the previous response express more empathic validation*) but found that this self-critique often changed generations too significantly; the model would tailor the generation to only address this tenet, ignoring the other tenets and much of the positive change it made in its first generation.

We further explored the optimal number of self-critiques. We found that after one self-critique, model generations either: 1) often sounded too arcane to be plausibly acceptable by students or teachers. Their rhetoric transitioned from realistic to very formulaic, following the tenets sometimes verbatim. Or 2) did not significantly change. Given these two outcomes, we decided that 1-critique was the optimal number of self-critiques for our setup.

### B.2 What about perturbations to the prompt?

We made small perturbations to the prompt: altering sentence structure, using synonyms, and reordering sentences. We did not find that any of these adjustments significantly changed the

model's outputs. We also tried these same perturbations on the instructions defining the model's task to reframe utterances in the system text and came to the same conclusion. We did not, however, try making any changes to the annotation guide (which was developed in a coordinated effort with expert teachers trained in GMSL language).

### B.3 What are the implications of these findings?

Our efforts in designing a GMSL-specific prompt indicate that GPT-4 is actually quite robust in its generations in GMSL contexts. But also too much context-specific information is a hindrance to accurate and realistic GMSL generations.

The model's strong performance in our evaluations came as a surprise to us—we expected the model to struggle in reframing utterances and did not anticipate students or teachers judging the models' utterances as better across all four of our metrics. Although the model reframings did perform well in our evaluations, we do acknowledge that ensuring the robustness and applicability of these generations across a wider variety of contexts still remains an open question that should be explored in future work. Also, understanding the reasoning behind LLMs' irregular behavior in these instances (some of which were described above) should also be the subject of future research.

## C Example of Complete Prompt Text

Below is the explicit, step-by-step process used to prompt the GPT-4 model. This is the text used in one example. The **System Text** and **Self-critique Prompt** remain constant for all examples. However, the **Example Prompt Text**, **Example Model Generation—Model (0-critiques)**, and **Example Model Generation—Model (1-critique)** change across examples.

### C.1 System Text

```
You are a helpful assistant who specializes in
    correcting teachers' language to be more
    supportive of the students. Your task is to
    directly revise the teacher's utterance to
    use mindset-supportive language.\n
Below are some tenets of mindset-supportive
    language to follow when encouraging students
    . Each tenet is followed by methods by which
     to accomplish that tenet. Adhere to these
    tenets and use mindset-supportive language
    when speaking with students.\n
1. Empathic validation: Thank students for
    asking question, Try to summarize the
    concern in new words, Compliment student on
```

bravery - reaching out is challenging and impressive. When possible, provide a legitimate, external cause for their concern (e.g. how society talks about college these days), rather than attributing it to them. If it is not possible or relevant to attribute their concerns to a legitimate external source, teachers should still attempt to validate emotions without blaming the student for having them. Never directly tell them they're wrong or mistaken.\n

2. Reappraisal of affect: Encourage them to view their negative emotions (fear, worry, stress, etc.) as a sign that they care deeply about something or someone, and reappraise the as resources that can empower them to take on the challenge.\n

3. Seek to Understand: Asking questions that will help you determine what the student has already done to approach the problem and what they are currently thinking or feeling.\n

4. Position yourself as a collaborative resource: Explain that you are trying to help them, Use terms like "we" or "us" to signal to the student that you are approaching a challenge with them.\n

5. Provide hope for change in the future (i.e. event-focused mindset, e.g. growth mindset): Explain how other people like them have been in a similar situation and have gone on to improve and be successful.\n

6. Autonomy-supportive language throughout: Try not to simply tell students what to do. Instead, try to offer students autonomy in their choices. You can use phrases like " have you considered" or "You can think of it in this way". The goal is not to tell students what to think or how to feel, but rather giving them the space to safely explore their thoughts and feelings about a problem.

## C.2 Example Prompt Text

```
This is an elementary grade math class working
    on fractions. They focus on with comparing
    fractions, with making equivalent fractions,
    simplifying fractions and putting fractions
    in order.\n
The following example of an in-class
    conversation between a teacher and student
    contains unsupportive language. Directly
    revise the teacher's last utterance, marked
    with "(Reframe)", to be more mindset-
    supportive:\n
Teacher: A pie. We don't use any of that in
    class obviously because we would all way too
    much if we were going to eat all that and
    it would spoil what do we talk about in our
    past class? A whole as being what?\n
Student: a fraction.\n
Teacher (Reframe): No, a fraction is part of the
    whole. What do we work with? I don't bring
    you candy bars. You guys don't bring in
    pizza. We talked about it. What do we use?
```

## C.3 Example Model Generation—MODEL (0-CRITIQUES)

```
Teacher (Revised): Thank you for sharing your
    thoughts. I appreciate your effort in
    remembering our past classes. A fraction is
    indeed a part of a whole. Let's think
    together, what do we usually use in class to
     represent a whole? It's not candy bars or
    pizza, but something else we've discussed
    before.
```

## C.4 Self-critique Prompt

```
Make the previous response more mindset
    supportive.
```

## C.5 Example Model Generation—MODEL (1-CRITIQUE)

```
Teacher (Revised): I appreciate your answer, and
     it's great to see you engaging with the
    concept of fractions. A fraction does
    represent a part of the whole. Now, let's
    work together to remember the examples we've
     used in class to showcase a whole. Remember
    , it wasn't food items like candy bars or
    pizza but something else we've talked about
    before. We'll figure it out together!
```

## D Examples of Original Utterances, Reframings, and Each Response Type's Evaluator Score

Below are five examples of conversations seen by student and teacher evaluators. The conversation excerpts include the **Classroom Context**, **Conversational Context**, and each of the four response types: ORIGINAL, EXPERT, MODEL (0-CRITIQUES), and MODEL (1-CRITIQUE). The scores given by both student and teacher evaluators to each of the responses are provided in Table 3. Scores were averaged across all student and teacher evaluations for a given response.[4]

We selected a diversely interesting set of examples to demonstrate that although model reframings often do perform the best, they are still imperfect; the specific cases in which models falter and the

---

[4]The scores in this table are averaged, raw scores. Here, *Perceived Teacher Mindset* and *Shame* are reverse coded: lower scores indicate better performance. A lower *Perceived Teacher Mindset* indicates the participant perceived the teacher to have more of a growth mindset whereas a higher score indicates the participant perceived the teacher to have more of a fixed mindset. A lower *Shame* score indicates the participant perceived the student as having less shame in the student-teacher interaction whereas a higher score indicates that the participant perceived the student as having more shame in the interaction.

divergence of student and teacher ratings should continue to be explored in the future.

**Conversation 1.** We see that EXPERT reframings outperform both model reframings in *Perceived Teacher Mindset* and *Shame* in student ratings and for every metric in teacher ratings.

**Conversation 2.** There is an inconsistency between student and teacher evaluations: in student evaluations, MODEL (0-CRITIQUES) outperforms MODEL (1-CRITIQUE), but this is reversed in teacher ratings as MODEL (1-CRITIQUE) at the top for all metrics.

**Conversation 3.** We again see that EXPERT reframings do very well, obtaining the best scores for ¾ of the student metrics. However, teachers rate MODEL (1-CRITIQUE) better across all three metrics.

**Conversation 4.** There is a large amount of inconsistency in the student ratings; Each reframing performs the best for at least one metric. In the teacher ratings, MODEL (0-CRITIQUES) obtains the highest ratings (tied in *Respect* with EXPERT).

**Conversation 5.** In this conversation, both the teacher and student raters marked MODEL (1-CRITIQUE) as the highest in *Shame*. Although MODEL (1-CRITIQUE) performed the best across all other metrics for both students and teachers, both teachers and students perceived it as inducing more shame than EXPERT, with students perceiving it as inducing the highest amount of shame amongst the four response types.

The variations in these results display the variability in quality for model outputs. Although, on average, and in our regression analysis, models perform better, they are not outperforming expert teachers' reframings in every scenario. Furthermore, we see that self-critique does not always improve upon generations—often the MODEL (0-CRITIQUES) reframings surpass MODEL (1-CRITIQUE) reframings. The pattern of situations in which self-critique hinders model performance should be explored in future work. These results also emphasize the need for future research to probe differences in preference between students and teachers in GMSL-specific contexts. As we can see, teacher and student evaluations sometimes differ. The settings in which students' and teachers' judgments diverge remain an open question that should be probed to truly understand the way in which GMSL can impact the classroom environment and students' learning outcomes.

## D.1 Conversation 1

**Classroom Context**

The teacher is talking about different kinds of triangles. Student H misclassified an isosceles triangle in class.

**Conversational Context**

Teacher: I think this side here is longer than these two. These two would be equal. So if two are equal, Student H, what's it called? Isosceles, scalene, or equilateral?
Student: Scalene.

**ORIGINAL**

Teacher: No.

**EXPERT**

Teacher: Thanks for sharing your answer! Let's think back about our root word in scalene. What does it mean? That everyone is different on the scale, right? So it can't be that one. Someone help Student H out.

**MODEL (0-CRITIQUES)**

Teacher: Thank you for your input, Student H. It's great that you're participating in the discussion. It can be a bit tricky to remember the different types of triangles. In this case, when two sides are equal, it's actually called an isosceles triangle. Keep engaging with the material, and we'll all get better at classifying triangles together.

**MODEL (1-CRITIQUE)**

Teacher: Thank you for sharing your thoughts, Student H. I appreciate your effort to answer the question. Classifying triangles can be a bit challenging at first. When two sides are equal, the triangle is called an isosceles triangle, not scalene. As we continue learning and practicing together, we'll become more confident in identifying these types of triangles. Remember, every mistake is an opportunity to learn and grow!

## D.2 Conversation 2

**Classroom Context**

This is an elementary grade math class working on fractions. They focus on locating fractions using a ruler as a number line.

**Conversational Context**

Teacher: How many lines [on a ruler] between zero and a half?
Student: Four.

**ORIGINAL**

Teacher: Point to them. Between zero and one half? How many lines is that in between?

**EXPERT**

Teacher: Thanks for sharing! How did you get four? Let's point to them on our ruler.

**MODEL (0-CRITIQUES)**

Teacher: Thank you for participating, and it's great that you are trying to figure this out. Let's work together to figure out how many lines are between zero and one half on the number line. Can you show me where you see the lines between zero and one half?

**MODEL (1-CRITIQUE)**

Teacher: I appreciate your effort in answering the question. Let's collaborate and take a closer look at the number line to figure out how many lines are between zero and one half. Can you help me identify those lines between zero and one half?

### D.3  Conversation 3

**Classroom Context**

This is an elementary grade math class working on long division.

**Conversational Context**

Teacher: Interesting. You made 6 groups of 6, and then you had 15 leftover. Is that what you said? I wonder, if these 15 could go in some of the other groups. Student: They can be reduced.

**ORIGINAL**

Teacher: Well I don't know if it would be reducing since it's not a fraction that we're making smaller, but that's a big number, 15. That number is larger than the number of groups that I have. So what that tells me is that I can take these 15 and still spread them out some more. You guys work on that.

**EXPERT**

Teacher: Oh, yes! That's a great way to put it, I like that you used that word. We usually only use it with fractions but I guess you can think about it like we are reducing our left-over pile! That's great thinking.

**MODEL (0-CRITIQUES)**

Teacher: Thank you for sharing that idea. It seems like you're thinking about reducing numbers, which is a great skill to have when working with fractions. In this situation, we're trying to spread the 15 equally among the groups we have. Let's think about how we can make that work. Can you come up with a way to distribute those 15 manipulatives among the existing groups?

**MODEL (1-CRITIQUE)**

Teacher: I appreciate your effort in trying to find a solution. It seems like you're applying the concept of reducing, which is an important technique when dealing with fractions. In this case, we want to focus on distributing the 15 evenly across the groups. Let's collaborate and explore some strategies to achieve this. What are your thoughts on how we can share those 15 manipulatives among the existing groups?

### D.4  Conversation 4

**Classroom Context**

This is an elementary grade class having a discussion about the relationship between multiplication and division. During the lesson they practice division math problems.

**Conversational Context**

Original Teacher: Where do I write that 2 for 12? Student: You don't put the 2 yet.

**ORIGINAL**

Teacher: Yeah we do.

**EXPERT**

Teacher: Interesting discussion about the 12 here! Let's go back and think about the 12. Once we multiple to get 12, how do we split the numbers?

**MODEL (0-CRITIQUES)**

Teacher: Thank you for sharing your thoughts. It can be a bit confusing at times. Let's work together to figure out where we should place the 2 for 12.

**MODEL (1-CRITIQUE)**

Teacher: Thank you for sharing your perspective. I understand that it might seem confusing at first. Let's collaborate and explore the correct placement of the 2 for 12, so we can learn together.

### D.5  Conversation 5

**Classroom Context**

This is an elementary grade math class working on geometry. They start the lesson with reviewing multiplication facts. They then begin to learn about classifying triangles focusing on types of angles.

**Conversational Context**

Original Teacher: Very good. Student A, on number six we have what in the corner? A little square. So is it a right triangle, or a different type? I can't hear you.

Student: Acute.

**ORIGINAL**

Teacher: No, sweetie. It's got a little square corner, so it makes a square. It's a right triangle. Now you know what I'm talking about, and do we think any sides are equal, Student E, in number six?

**EXPERT**

Teacher: Thanks for sharing your answer! Let's

remember what our square corner tells us about triangles.

**MODEL (0-CRITIQUES)**

Teacher: Thank you for your response, Student A. When we see a little square in the corner, it actually represents a right angle. So, this would be a right triangle. It can be confusing sometimes, but now we know what to look for. As for the sides, Student E, what do you think about their lengths in number six?

**MODEL (1-CRITIQUE)**

Teacher: I appreciate your input, Student A. When there's a little square in the corner, it signifies a right angle. That means this is a right triangle. It's completely understandable to mix them up, but now we understand the difference. Moving on to the sides, Student E, do you have any thoughts about their lengths in number six?

## E    Additional Information on Surveys

Below, we provide an example of survey frames seen by participants. These figures provide an overview of the survey experience for participants. The frames we include are: the consent form (Figure 4), the survey overview (Figure 5), an example conversation transcript (Figure 6), example questions (Figure 7 and Figure 8). We also include an example of the complete survey in the Supplementary Materials.

| Conversation | Response Type | Student Ratings | | | | Teacher Ratings | | |
|---|---|---|---|---|---|---|---|---|
| | | **Perceived Teacher Mindset** | **Challenge-Seeking** | **Shame** | **Respect** | **Perceived Teacher Mindset** | **Shame** | **Respect** |
| 1 | ORIGINAL | 4.27 | 1.27 | 5.09 | 3.46 | 3 | 4.5 | 4 |
| | EXPERT | **1.95** | 1.5 | **3** | 5.1 | **2.75** | **3** | **5.5** |
| | MODEL (0-CRITIQUES) | 2.82 | **1.55** | 3.27 | **5.18** | 4* | 4* | 3* |
| | MODEL (1-CRITIQUE) | 2.61 | 1.22 | 3.44 | 4.78 | 2* | 4* | 5* |
| 2 | ORIGINAL | 3.46 | 1.33 | 4.17 | 4.33 | 3 | 4 | 4.5 |
| | EXPERT | 3.2 | 1.3 | 4.2 | 4.7 | 2.5* | 3* | 5* |
| | MODEL (0-CRITIQUES) | **2.4** | **1.4** | **2.5** | **5.4** | 2.5 | 3.5 | 4.5 |
| | MODEL (1-CRITIQUE) | 2.95 | 1.3 | 4.1 | 5.1 | **2.17** | **2.33** | **5.33** |
| 3 | ORIGINAL | 3.08 | 1.08 | 4.67 | 4.42 | 2.83 | 3.67 | 4.67 |
| | EXPERT | **2.45** | **1.8** | **2.9** | 4.9 | 2* | 3* | 5* |
| | MODEL (0-CRITIQUES) | 2.95 | 1.6 | 3.4 | 4.7 | 3.5 | 3 | 4.5 |
| | MODEL (1-CRITIQUE) | 3.82 | 1.36 | 3.91 | **4.91** | **1.75** | **2.5** | **6** |
| 4 | ORIGINAL | 3.55 | 1 | 5 | 2.55 | 3* | 4* | 4* |
| | EXPERT | 3.23 | 1.4 | **3.18** | 4.3 | 1.75 | 4.5 | **5** |
| | MODEL (0-CRITIQUES) | **2.82** | 1.45 | 3.27 | 4.64 | **1.25** | **2** | **5** |
| | MODEL (1-CRITIQUE) | 2.85 | **1.5** | 3.6 | **5.4** | 2* | 3* | 4* |
| 5 | ORIGINAL | 2.85 | 1.1 | 3.9 | 4.4 | 4.17 | 5 | 3.3 |
| | EXPERT | 2.59 | 1.27 | 3.45 | **4.45** | **1.75** | **2.5** | **5** |
| | MODEL (0-CRITIQUES) | **2.55** | **1.3** | **2.9** | 4.3 | 2.5 | 3 | 4.5 |
| | MODEL (1-CRITIQUE) | 3.72 | 1.09 | 3.9 | 4 | 4 | 5 | 3 |

Table 3: Conversations' evaluator scores for each response type. * denotes that only one evaluator rated the response.

**Basic Study Information**

The purpose of the study is to investigate educational practices and ways to improve educational materials. We are conducting this study with adults more generally (i.e., not just teachers) in order to obtain broad feedback on the clarity, persuasiveness, relevance, and effectiveness of our materials.

If you agree to take part in this study, you will be asked to complete an online survey in which you will do the following.
- Answer the questions provided in the online survey.
- At the end, you will complete some demographics questions.

**How long will this study take and how many people will be in the study?**

Completion of the online survey will take approximately 10-15 minutes. Approximately 250 U.S. adults will participate in this study.

**What risks and discomforts might you experience from being in this study?**

To the best of our knowledge, the tasks you will be completing have no more risk of harm to you than what you would experience in everyday life. There may be a risk of boredom, fatigue or discomfort while completing the questionnaire. Participation is completely voluntary and may be terminated at any time.

**How could you benefit from this study?**

You will likely receive no direct benefit from participating in this study. However, this research may contribute to improved educational processes in high schools and colleges across the country.

**How will we protect your information?**

We will collect your email only to faccillitate payment. Any presentations or papers published about the findings from this study will therefore not include your name or any other information that could identify you. We may share the data we collect in this study with other researchers for future research studies – if we share your data, we will not include any information that can identify you.

**How will we compensate you for being part of the study?**

Your compensation for completing this study will be provided through email. Please note partial compensation is not offered; you must complete the entire study to receive the compensation points. Compensation will be made in the amount of $15.00 to be given to you via an e-gift card as an Amazon e-gift card delivered to your email address.

**Your Participation in this Study is Voluntary**

Participation in this study is voluntary. If at any time and for any reason, you would prefer not to participate in this study, you may stop at any time by closing the browser window to withdraw from the study without penalty.

**Contact Information for the Study Team and Questions about the Research**

If you have any questions about this research, you may contact the research team contact:

    Anonymized

**Contact Information for Questions about Your Rights as a Research Participant**

If you have questions about your rights as a research participant, or wish to obtain information, ask questions, or discuss any concerns about this study with someone other than the researcher(s), please contact the following:

Institutional Review Board

    Anonymized

Please reference the protocol number found at the top of this document.

**Your Consent**

By clicking the button below to continue to the next page, you are indicating that you reside in the United States, are at least 18 years old, have read and understand this consent form, and you agree to participate in this online research study. If at any time you wish to stop participating, simply close your browser window.

| Yes, I consent to taking this survey |
|---|

| No, I do not consent to taking this survey |
|---|

Figure 4: Consent required for all participants as described in the **Ethics Statement**.

## Overview of This Activity

As a student, you are in a position to provide teachers with helpful feedback about what they are (and aren't) doing well to help you learn. Unfortunately, teachers rarely receive this important feedback from their students.

Today, we'll ask you to read an example of what four math teachers said during their classes, and we'll ask you to tell us what you think of these teachers. Your responses will be helpful for training future teachers so that they know how to support students like you.

This survey includes opinion questions about what teachers say to their classes. Please do your best to give it your complete attention and answer all questions honestly.

→

Figure 5: Survey overview shown to participants in Section 4.1.

In this transcript, the teacher is telling the students about different multiplication properties. The associate property of multiplication states that when multiplying three numbers together, regardless of how they're grouped, the answer will always be the same. For example, (3 x 4) x 5 = 60 and 3 x (4 x 5) = 60.

---

**Transcript:**

Teacher: And today's word means when I have a math problem and it has parentheses, and I choose to change the problem a little bit by moving the parenthesis, and it's still equal I've got a problem. Everybody put on your thinking caps. I'll give you a clue. It's in the first column. Student A?

Student A: It's the distributive property.

Teacher: Why do you think that is the answer? Tell me what you are thinking.

---

Figure 6: Example conversation transcript shown to participants in Section 4.1.

Figure 7: Example question shown to participants in Section 4.1 to understand participants' perception of hypothetical teacher's growth mindset.

Figure 8: Example question shown to participants in Section 4.1 to understand participants' willingness to engage in challenge-seeking behavior if they were in this hypothetical teacher's class.