# OpenReview forum: "“Mistakes Help Us Grow”: Facilitating and Evaluating Growth Mindset Supportive Language in Classrooms"
_EMNLP/2023/Conference — EMNLP 2023 Main_

### Official Review · Reviewer_swnb · 2023-08-04

**Soundness:** 3

**Excitement:**

3: Ambivalent: It has merits (e.g., it reports state-of-the-art results, the idea is nice), but there are key weaknesses (e.g., it describes incremental work), and it can significantly benefit from another round of revision. However, I won't object to accepting it if my co-reviewers champion it.

**Paper Topic And Main Contributions:**

The paper investigates whether GPT4 is able to generate growth mindset supportive language (GMSL) reframings on utterances made, originally, by teachers in a contextual classroom environment. The paper designs a new taxonomy to annotate GMSL language, uses it to prompt engineer GPT4, and evaluates the perceived impact of these reframings (and other actual expert-made reframings) through teacher and student feedback and lexical analysis.

Model responses are shown to be greatly preferred against original teacher responses, and elicit challenge-seeking behavior—a desired GMSL outcome—in students, measured on a likert scale using survey responses. The lexical analysis of n-grams also shows some features of high GMSL considered language, which matches intuition.

**Reasons To Accept:**

Overall, a very interesting study on how large language models can be used to encourage positive outcomes in educational settings, with some very surprising results—LLM reframings even outperformed expert reframings! The key reason to accept here is the novel domain paired with actual human evaluation, which makes this paper stand out among its peers.

**Reasons To Reject:**

A key concern I have with the paper is in its evaluation: the authors use hypothetical survey questions to ask and measure how students and teachers would choose to act. Using this method of evaluation is quick and easy, but the key challenge remains to see if these results would hold in an actual classroom environment and impact *actual student behavior and learning outcomes.* This seems especially pertinent in the education domain, particularly when the paper's key novelty is in this unique setting compared to existing prior work in positive reframing (i.e. Ziems 2022, using social media data)—which, without, makes the key evaluation in this paper that of one on *perception* than actual educational impacts. Establishing this link between hypothetical student scenarios and actual learning outcomes is key to making this a stronger paper.

**After author response:** Thanks to the authors for their response; raising soundness from 2 to 3.

**Reproducibility:**

5: Could easily reproduce the results.

**Reviewer Confidence:**

5: Positive that my evaluation is correct. I read the paper very carefully and I am very familiar with related work.

---

> ### Author Rebuttal · Authors · 2023-08-29
>
> We thank the reviewer for their feedback! We are happy they appreciate the potential of our work to use LLMs to “encourage positive outcomes in educational settings, with some very surprising results.” **We will revise the paper to better motivate our evaluation setup and goals for future work.**
>
> > Reasons to reject: A key concern I have with the paper is in its evaluation: the authors use hypothetical survey questions to ask and measure how students and teachers would choose to act. Using this method of evaluation is quick and easy, but the key challenge remains to see if these results would hold in an actual classroom environment and impact actual student behavior and learning outcomes. This seems especially pertinent in the education domain, particularly when the paper's key novelty is in this unique setting compared to existing prior work in positive reframing (i.e. Ziems 2022, using social media data)—which, without, makes the key evaluation in this paper that of one on perception than actual educational impacts. Establishing this link between hypothetical student scenarios and actual learning outcomes is key to making this a stronger paper.
>
> We agree with the reviewer that it is critical to evaluate the effects of GMSL on actual student learning. However, before this could be done, it was essential to examine LLM’s abilities to generate language that is perceived as GMSL. **As a result, the goal of our paper is to establish the validity of applying LLMs in instances of unsupportive language. Our positive results then can motivate future longitudinal studies deploying LLM-based tools to provide GMSL feedback in classrooms.**
> Prior work, through long-term longitudinal studies, has demonstrated the correlation between GMSL intervention and student learning outcomes across metrics such as grades, higher course passing rates, and selection of advanced mathematics courses (Yeager et al., 2019; Rege et al., 2021). These results were consistent across achievement levels and countries (OECD, 2019). Prior research has also shown the necessity of a teachers’ growth-mindset, communicated through their supportive language, to allow students to profit from their own growth-mindsets (Yeager et al., 2021). Furthermore, **follow-up studies have also demonstrated that students’ hypothetical judgments of teachers’ mindset supportive language did track with teachers’ actual behavior and students’ learning outcomes (Hecht et al., 2022).** Our research builds off of this understanding.
>
> **Our work is an important contribution because it is the first study, to our knowledge, that explores LLMs’ ability to utilize GMSL.** Classroom evaluations are long-term, logistically challenging, and expensive; we aim to first establish the feasibility of LLM’s deployment for this task. Through our teacher evaluations, we identify that teachers are receptive to LLM reframed responses. This is key to our overarching goal of supporting teachers’ instruction in classrooms and a precondition to an eventual longer, longitudinal study. Additionally, prior work demonstrates a disconnect between student perception of teachers’ utterances and teachers’ perception of their own language, leading to learning outcomes which differ from student and teacher learning goals (Dweck, 2015; Barger, Xiong, & Ferster 2022). Thus, our extensive student evaluation is necessary to establish that the GMSL reframings by LLMs are received positively by the students who are on the receiving end of the language.
>
> Having identified LLM’s potential in this domain, the reviewer’s proposition is our ultimate goal—identifying whether these reframings impact students' learning outcomes within the actual classroom environment. **We follow a precedent for an interactive evaluation setup in education and computer science literature.** For example, Demszky et al. (2021) explored a computational framework for measuring conversational uptake (in ACL 2021) prior to conducting a large-scale experiment—Demszky et al. (2023)—in which they deployed their previously established framework in an actual classroom environment (journal article in Educational Evaluation and Policy Analysis). Their initial paper served as an early validation of their overall framework and approach prior to conducting the in-practice experiment. Our research mirrors this approach.
>
> Yeager, D.S., Hanselman, P., Walton, G.M. et al. A national experiment reveals where a growth mindset improves achievement. Nature 573, 364–369 (2019). https://doi.org/10.1038/s41586-019-1466-y
>
> Rege, M., Hanselman, P., Solli, I. F., Dweck, C. S., Ludvigsen, S., Bettinger, E., Crosnoe, R., Muller, C., Walton, G., Duckworth, A., & Yeager, D. S. (2021). How can we inspire nations of learners? An investigation of growth mindset and challenge-seeking in two countries. American Psychologist, 76(5), 755–767. https://doi.org/10.1037/amp0000647
>
> Yeager, D. S., Carroll, J. M., Buontempo, J., Cimpian, A., Woody, S., Crosnoe, R., Muller, C., Murray, J., Mhatre, P., Kersting, N., Hulleman, C., Kudym, M., Murphy, M., Duckworth, A. L., Walton, G. M., & Dweck, C. S. (2022). Teacher Mindsets Help Explain Where a Growth-Mindset Intervention Does and Doesn’t Work. Psychological Science, 33(1), 18–32. https://doi.org/10.1177/09567976211028984
>
> Demszky, D., Liu, J., Mancenido, Z., Cohen, J., Hill, H., Jurafsky, D., & Hashimoto, T. (2021). Measuring conversational uptake: A case study on student-teacher interactions. In Proceedings of the 59th Annual Meeting of the Association for Computational Linguistics and the 11th International Joint Conference on Natural Language Processing (Volume 1: Long Papers) (pp. 1638–1653). Association for Computational Linguistics. https://doi.org/10.18653/v1/2021.acl-long.130
>
> Demszky, D., Liu, J., Hill, H. C., Jurafsky, D., & Piech, C. (2023). Can Automated Feedback Improve Teachers’ Uptake of Student Ideas? Evidence From a Randomized Controlled Trial in a Large-Scale Online Course. Educational Evaluation and Policy Analysis, 0(0). https://doi.org/10.3102/01623737231169270
>
> Hecht, C. A., Dweck, C. S., Murphy, M. C., Kroeper, K. M., &amp; Yeager, D. S. (2022). Efficiently exploring the causal role of contextual moderators in behavioral science. Proceedings of the National Academy of Sciences, 120(1). https://doi.org/10.1073/pnas.2216315120
>
> OECD. (2019). PISA 2018 results (Volume III): What school life means for students’ lives. PISA, OECD Publishing. https://doi.org/10.1787/acd78851-en
>
> Barger, M. M., Xiong, Y., Ferster, A. E. (2022). Identifying false growth mindsets in adults and implications for mathematics motivation. Contemporary Educational Psychology, 70, 102079. https://doi.org/10.1016/j.cedpsych.2022.102079
>
> Dweck, C. (2015). Carol Dweck Revisits the “Growth Mindset.” Student Well-Being. August 27, 2023, https://www.edweek.org/leadership/opinion-carol-dweck-revisits-the-growth-mindset/2015/09

---

### Official Review · Reviewer_QY3p · 2023-08-04

**Soundness:** 4

**Ethical Concerns:**

Yes

**Excitement:**

4: Strong: This paper deepens the understanding of some phenomenon or lowers the barriers to an existing research direction.

**Justification For Ethical Concerns:**

Difficult to judge with just one paper if there are no cultural aspects to be considered here -- Marshmallow Effect

**Paper Topic And Main Contributions:**


## Summary

The work consider utterances of teacher T -- student S dialogues in math problem solving context. Keeping a Growth Mindset Supportive Language (GMSL) approach, the paper could be used to promote T-S dialogue in positive directions. In several instances, in the heat of the moment, T may provide adverse feedback to S which could lead to S's lack of motivation, lack of engagement in the task and higher chance of dropouts. In summary, GMSL is a technique to promote T-S collaboration towards common goals. The work boils down to the partial automation of this process with the use of LLMs, in collaboration with teachers trained in GMSL.

**Questions For The Authors:**

1. How does contextual definition of GMSL change, by subject, by teacher -- student pair, level of expertise of student as well instructor?


2. How do the specificities of the dataset affect the automated model of GMSL generation step, approximately in how many cases was teacher trained in GMSL intervention necessary?

**Reasons To Accept:**


## Reasons

1. Strong factors that influence student's learning outcomes are considered. Promoting this work would effectively promote dialogue  and interest in the directions of student mental learning and its impact on learning outcomes.

2. This work has long-term societal impacts with the use of LLMs in positive directions, which could help drive away some negative notions developed around generative AI as being harmful because of its capabilities. This is a timely work.

## Strengths

1. Difficult to measure qualitative aspects of teacher -- student dialogue, one may consider growth mindset, others may consider it as harsh feedback. This work attempts at nailing this aspect down in the form of a dataset with 2 levels of critique (0-moderated 1 - moderated) by teachers trained in GMSL.

2. The paper plays an exemplary role of human-ai collaboration in a difficult use that could impact learning outcomes.

**Reasons To Reject:**


## Reasons

1. Some learning outcomes are difficult to segregate as learning units. For example, when student attempts to learn concept in math, there are some intermediate steps involved towards the learning goals

Refer --
Koedinger, Kenneth R., and Mitchell J. Nathan. "The real story behind story problems: Effects of representations on quantitative reasoning." The journal of the learning sciences 13.2 (2004): 129-164.

Lovett (1998). Cognitive task analysis in service of intelligent tutoring system design.
In Proceedings of the Intelligent Tutoring Systems Conference.

Koedinger, Kenneth R., Albert T. Corbett, and Charles Perfetti. "The Knowledge‐Learning‐Instruction framework: Bridging the science‐practice chasm to enhance robust student learning." Cognitive science 36.5 (2012): 757-798.

How can this be modeled with growth mindset better, so that student - teacher doesn't deviate from the broader learning outcomeS?

## Weaknesses

1. Difficult to quantify learning outcomes, when quantifying student - teacher dialogue. A Teacher may provide harsh feedback at times so incite student to think in the correct direction. This could probably work in cases of some students without adverse impact on the student pysche. Providing feedback is correlated with the learning outcomes


**Reproducibility:**

3: Could reproduce the results with some difficulty. The settings of parameters are underspecified or subjectively determined; the training/evaluation data are not widely available.

**Reviewer Confidence:**

5: Positive that my evaluation is correct. I read the paper very carefully and I am very familiar with related work.

---

> ### Author Rebuttal · Authors · 2023-08-29
>
> We thank the reviewer for their feedback! We are happy they appreciate our work’s “long-term societal impacts with the use of LLMs in positive directions” and value its “exemplary role of human-ai collaboration in a difficult use that could impact learning outcomes.” **We will revise the paper to include more context of prior work that establishes the connection between student-teacher dialogue and learning outcomes. We will also add details on the need for future work to explore the cultural variables that might impact GMSL.**
>
> > Reasons 1: Some learning outcomes are difficult to segregate as learning units. For example, when student attempts to learn concept in math, there are some intermediate steps involved towards the learning goals. How can this be modeled with growth mindset better, so that student - teacher doesn't deviate from the broader learning outcome?
>
> **Because our reframings are performed on a sentence-level in context of the existing conversation, we only alter the way the learning goals are communicated and not the learning goals themselves.** Prior work has demonstrated that these sentence-level changes of GMSL improve student learning outcomes while remaining within the bounds of the class’ overall goals (Yeager et al., 2019). When initially prompt-tuning GPT-4 for the reframing task, we did not include significant conversational context. In these initial experiments, we did notice that the models sometimes struggled to maintain the central learning goal within the conversation. Our final prompting-tuning setup originated from this initial problem—we found that by providing both the classroom context, which often represents the long term learning goals (*“This is an elementary grade math class working on geometry. They start the lesson with reviewing multiplication facts. They then begin to learn about classifying triangles focusing on types of angles.”*) and the two utterances directly prior, GPT-4 was able to maintain the central topic of discussion and focus only on reframing. GPT-4 reframings maintained the same scope as the teacher reframings (thereby not straying from the learning goals) while improving upon the actual language within the reframing.
>
> In future work, we hope to investigate the long-term outcomes associated with GPT-4 reframings—an unexplored but potentially impactful domain—demonstrating LLMs’ to enhance educators’ teaching experience and students’ learning outcomes in the long run.
>
> Yeager, D.S., Hanselman, P., Walton, G.M. et al. A national experiment reveals where a growth mindset improves achievement. Nature 573, 364–369 (2019). https://doi.org/10.1038/s41586-019-1466-y
>
> > Weaknesses 1: Difficult to quantify learning outcomes, when quantifying student - teacher dialogue. A Teacher may provide harsh feedback at times so incite student to think in the correct direction. This could probably work in cases of some students without adverse impact on the student pysche. Providing feedback is correlated with the learning outcomes
>
> Although we recognize that quantifying learning outcomes through student-teacher dialogue is difficult, recent work has shown a causal relationship between students’ perceptive judgments of teachers’ mindset supportive language and teachers’ in-class behavior as well as students’ learning outcomes (Hecht et al. 2022). Previous research of GMSL has also demonstrated the long-term impact of GMSL-intervention within classroom contexts; for example, Yeager et al. (2019) found that exposing students to a growth mindset in a short one-hour conversation improved the grades of lower-achieving students and increased overall enrollment to advanced mathematics courses (see also Schmidt et al. (2017) and Muenks et al. (2022) for similar findings).
>
> Thus, GMSL builds upon student’s learning goals rather than altering or overshadowing them. **We build upon this understanding to explore automating GMSL feedback so that any teacher can integrate GMSL into their instruction; currently the lack of effective GMSL coaching restricts teachers’ implementation of GMSL.** GMSL **does not** advocate for the lack of providing feedback but rather restructures the way in which the feedback is provided. GMSL feedback uses language that focuses on teaching that intellectual abilities can be developed. It does not necessitate that the feedback is not harsh or constructive, but rather that it communicates that growth is possible for the student. For instance, the following is an example from the NCTE transcript:
>
> *Teacher: Mm-hmm, so erase your fraction and put one. Okay. You’re still wrong. So zero and one. How many pieces is that?*
>
> *Student: Three.*
>
> *Teacher: Erase what you have please. Everything. And I want you to look at it again, okay? Everything. Everything. Everything.*
>
> Here, the teacher is correcting a student’s mistake. They are offering criticism, telling the student that they are wrong, and asking them to try again. However, the language used is unsupportive. Our expert teachers reframed this utterance to:
>
> *Teacher: That’s a great answer. But let’s look again. We’re going to start with a clean slate and see if that helps. Let’s erase everything.*
>
> The reframing does not remove criticism/feedback from the teacher’s response but rather restructures the response to be more growth-oriented, improving students’ overall learning process. It is not that we want to avoid providing feedback, but rather that the feedback should be provided in such a way that students’ can best improve their learning process by way of that feedback (more examples of conversations from the NCTE dataset and their reframings are in Section C of the Appendix). The primary focus of our current paper is to explore, assuming that the growth mindset approach is generally accepted as beneficial for teachers and students, what contributions we can make from a technical standpoint.
>
> Yeager, D.S., Hanselman, P., Walton, G.M. et al. A national experiment reveals where a growth mindset improves achievement. Nature 573, 364–369 (2019). https://doi.org/10.1038/s41586-019-1466-y
>
> Schmidt, J. A., Shumow, L., & Kackar-Cam, H. Z. (2017). Does Mindset Intervention Predict Students' Daily Experience in Classrooms? A Comparison of Seventh and Ninth Graders' Trajectories. Journal of youth and adolescence, 46(3), 582–602. https://doi.org/10.1007/s10964-016-0489-z
>
> Muenks, K., Canning, E. A., LaCosse, J., Green, D. J., Zirkel, S., Garcia, J. A., & Murphy, M. C. (2020). Does my professor think my ability can change? Students’ perceptions of their STEM professors’ mindset beliefs predict their psychological vulnerability, engagement, and performance in class. Journal of Experimental Psychology: General, 149(11), 2119–2144. https://doi.org/10.1037/xge0000763
>
> > Question 1: How does contextual definition of GMSL change, by subject, by teacher -- student pair, level of expertise of student as well instructor?
>
> While the tenets presented in the GMSL guide are applicable to a variety of contexts (e.g. empathetic validation, expressing hope for change), the **specific language** for instantiating these tenets may indeed vary by context. For example, for both a high school mathematics context and a fourth-grade context, the educator should express to students that they believe in the student’s ability to succeed despite potential difficulties. However, the language used between the highschool course and the fourth-grade classroom will be different. To use GMSL in a highschool context, the educator might say, *“I believe you are all capable of doing well in this class, even when you face difficulties.”* Alternatively, when talking to a fourth-grade classroom, the educator might say, *“I know that you are all going to be great math learners by the end of the year, even when you might think it’s hard.”* These express the same idea, and both use GMSL, but are necessarily contextually different. As our work is the first to explicate the high-level tenets of GMSL and translate it to teachers’ discourse, we decided to work with a concrete demographic—4th-6th grade math classrooms—to create a well-defined scope for annotation and evaluation, and to reduce sparsity in our resulting dataset. Working with teachers to adapt our GMSL guide to other subjects, grade-levels and student’s learning needs is a promising direction for future work.
>
> > Question 2: How do the specificities of the dataset affect the automated model of GMSL generation step, approximately in how many cases was teacher trained in GMSL intervention necessary?
>
> Model generations were very sensitive to the classroom context (grade level, subject) provided in the prompt (Section B of the Appendix). For example, in prior prompt-tuning setups that did not include the class year, the model generations contained fewer examples, and the language was more rigid. However, the overall content/information communicated did not change significantly. But, as detailed in our earlier response, if the classroom context was missing altogether, the model often strayed from the classroom’s overall learning goal, often adding in tangentially related information or widening the scope of the issue the conversation was attempting to address. We did not notice significant differences in reframing strategies employed by the teachers trained in GMSL or GPT-4 across different student-teacher pairs. That being said, we encourage future work to explore the implications of these variables in LLMs’ generation of GMSL.
>
> Across the transcripts used for prompt-tuning and testing the GPT-4 model, 22.8% of all teacher utterances (100 out of 438 teacher utterances) required reframing (in the paper we mistakenly reported the rate to be 17%, which we will update in the revision).
>
> > Ethics Concerns: Difficult to judge with just one paper if there are no cultural aspects to be considered here -- Marshmallow Effect
>
> As our paper is the first, to our knowledge, to both make explicit the linguistic properties of GMSL and demonstrate the use of LLMs to provide GMSL in classroom contexts, we hope to start the discussion regarding various cultural aspects that impact the utility of GMSL for students and teachers. The evaluation framework we propose in this paper can hopefully lead to future research in this area, tackling other variables, such as cultural aspects, that might impact students’ and teachers’ treatment of GMSL. We will revise our paper to elaborate on this constraint in the ethics and limitations sections.

---

### Official Review · Reviewer_jpn8 · 2023-08-05

**Typos Grammar Style And Presentation Improvements:** None
**Soundness:** 4

**Excitement:**

4: Strong: This paper deepens the understanding of some phenomenon or lowers the barriers to an existing research direction.

**Missing References:**

None

**Paper Topic And Main Contributions:**

This paper presented an interesting study of using large language models to provide automated coaching to support teachers' use of growth mindset supportive language. Specifically, the paper made three contributions, including (1) an annotation framework to guide teacher's understanding of growth mindset supportive language; (2) an open-source dataset containing both unsupportive and supportive utterances guided by the defined annotation framework; and (3) an evaluation framework to assess the perceived impact of growth mindset supportive language in education.

**Questions For The Authors:**

1. Could the authors describe the process of developing the annotation guide in more details to increase the validity of the developed framework? e.g., how was the initial GMSL Guide created and based on any source transcripts? Did the expert teachers identified the important suggestions separately and then reconvened to discuss and resolve disagreement (if any)?

**Reasons To Accept:**

1. This is the first study attempting to use large language models to provide growth mindset supportive language in education.
2. The annotation framework of growth mindset supportive language can be of use to other relevant researchers in the field.
3. The empirical results regarding the perceived impact of growth mindset supportive language are insightful.

**Reasons To Reject:**

None

**Reproducibility:**

4: Could mostly reproduce the results, but there may be some variation because of sample variance or minor variations in their interpretation of the protocol or method.

**Reviewer Confidence:**

4: Quite sure. I tried to check the important points carefully. It's unlikely, though conceivable, that I missed something that should affect my ratings.

---

> ### Author Rebuttal · Authors · 2023-08-29
>
> We thank the reviewer for their feedback! We are glad they find our empirical results insightful, value the usefulness of our annotation framework, and appreciate the novelty of our use of LLMs to provide GMSL. **We will revise the paper to add further detail about the process of developing the GMSL Guide.**
>
> > Question 1: Could the authors describe the process of developing the annotation guide in more details to increase the validity of the developed framework? e.g., how was the initial GMSL Guide created and based on any source transcripts? Did the expert teachers identified the important suggestions separately and then reconvened to discuss and resolve disagreement (if any)?
>
> The GMSL Guide is developed in collaboration with growth mindset experts and elementary school teachers trained in GMSL. The process was as follows:
> 1. GMSL experts created an initial draft of a GMSL guide. This initial draft was informed by prior research highlighting the importance of educators' usage of GMSL (Yeager et al., 2021). The GMSL experts then met with the teachers to review the guide, review additional resources on general growth mindset information, and to discuss the internal thought process of identifying unsupportive language and how to reframe those instances into GMSL.
> 2. The teachers were then provided three practice NCTE transcripts and were asked to independently identify utterances needing reframing and to reframe those into GMSL. Anecdotally, many of the utterances selected for reframing were the same between teachers. These initial three transcripts were not used in our test set. During this phase, the teachers were also asked to document their annotation process.
> 3. Then, the GMSL experts met with the teachers to discuss their annotation process and GMSL reframing. In these meetings, we synthesized their experience of the annotation process to create an updated version of the GMSL Guide. During these conversations, we did not notice significant disagreement between the teachers.
> 4. After developing the more finalized version of the GMSL guide, the teachers then independently annotated the eight transcripts that were used in the test set. These final reframings were the ones used to prompt-tune and test GPT-4.
>
> Yeager, D. S., Carroll, J. M., Buontempo, J., Cimpian, A., Woody, S., Crosnoe, R., Muller, C., Murray, J., Mhatre, P., Kersting, N., Hulleman, C., Kudym, M., Murphy, M., Duckworth, A. L., Walton, G. M., & Dweck, C. S. (2021). Teacher mindsets help explain where a growth-mindset intervention does and doesn’t work. Psychological Science, 33(1), 18–32. https://doi.org/10.1177/09567976211028984

---

### Meta-Review · Area_Chair_6aqZ · 2023-09-16

**Recommendation:** 4

**Metareview:**

The paper explores how LLMs can be used to support teachers in using growth mindset supportive language (GMSL). Teachers annotate classroom transcripts and provide expert reframings. GPT-4 is used to reframe student-directed language to exhibit a growth mindset. Then, student and teachers are surveyed on (1) teacher’s perceived growth mindset, (2) teacher’s promotion of challenge-taking behavior, (3) shame felt by students in the teacher’s class, and (4) respect. It is found that GPT-4 responses are often scored higher than expert responses.

This paper displays a nice application of LLMs with positive social impact. The issues mentioned by reviewer jpn8 regarding the GMSL guide seem to have largely been addressed. Reviewer swnb's point regarding testing this approach in a classroom is an important one, but the paper can stand alone prior to such testing (with the expectation of future publication of time-consuming follow-up work). This also is related to reviewer QY3p's point about actual learning outcomes.

---

### Decision · Program_Chairs · 2023-10-07

**Decision:**

Accept-Main

**Comment:**

The paper explores how LLMs can be used to support teachers in using growth mindset supportive language (GMSL). Teachers annotate classroom transcripts and provide expert reframings. GPT-4 is used to reframe student-directed language to exhibit a growth mindset. Then, student and teachers are surveyed on (1) teacher’s perceived growth mindset, (2) teacher’s promotion of challenge-taking behavior, (3) shame felt by students in the teacher’s class, and (4) respect. It is found that GPT-4 responses are often scored higher than expert responses.

This paper displays a nice application of LLMs with positive social impact. The issues mentioned by reviewer jpn8 regarding the GMSL guide seem to have largely been addressed. Reviewer swnb's point regarding testing this approach in a classroom is an important one, but the paper can stand alone prior to such testing (with the expectation of future publication of time-consuming follow-up work). This also is related to reviewer QY3p's point about actual learning outcomes.